# STORYAGENT: CUSTOMIZED STORYTELLING VIDEO GENERATION VIA MULTI-AGENT COLLABORATION

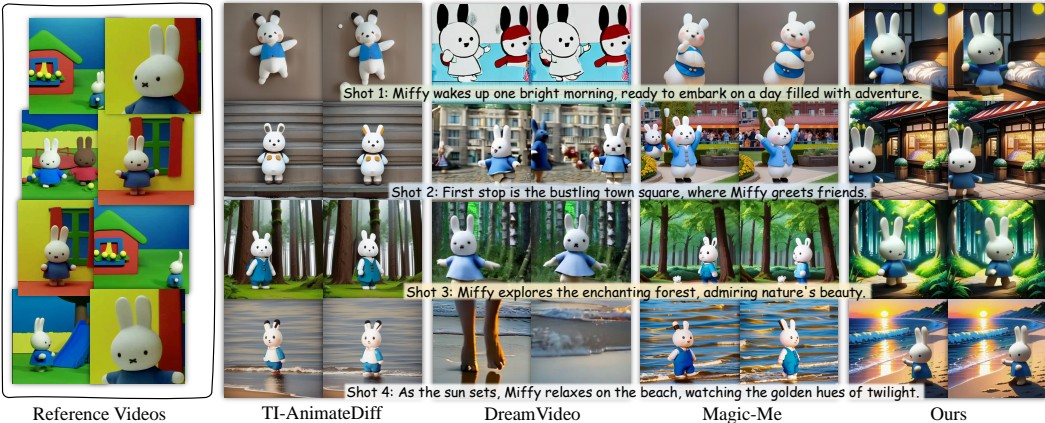

Figure 1: Comparison results of customized storytelling videos. Existing methods fail to preserve the subject consistency across shots, while our method successfully maintains inter-shot and intra-shot consistency of the customized subject.

## ABSTRACT

The advent of AI-Generated Content (AIGC) has spurred research into automated video generation to streamline conventional processes. However, automating storytelling video production, particularly for customized narratives, remains challenging due to the complexity of maintaining subject consistency across shots. While existing approaches like Mora and AesopAgent integrate multiple agents for Story-to-Video (S2V) generation, they fall short in preserving protagonist consistency and supporting Customized Storytelling Video Generation (CSVG). To address these limitations, we propose StoryAgent, a multi-agent framework designed for CSVG. StoryAgent decomposes CSVG into distinct subtasks assigned to specialized agents, mirroring the professional production process. Notably, our framework includes agents for story design, storyboard generation, video creation, agent coordination, and result evaluation. Leveraging the strengths of different models, StoryAgent enhances control over the generation process, significantly improving character consistency. Specifically, we introduce a customized Image-to-Video (I2V) method, LoRA-BE, to enhance intra-shot temporal consistency, while a novel storyboard generation pipeline is proposed to maintain subject consistency across shots. Extensive experiments demonstrate the effectiveness of our approach in synthesizing highly consistent storytelling videos, outperforming state-of-the-art methods. Our contributions include the introduction of StoryAgent, a versatile framework for video generation tasks, and novel techniques for preserving protagonist consistency.

## 1 INTRODUCTION

Storytelling videos, typically multi-shot sequences depicting a consistent subject such as a human, animal, or cartoon character, are extensively used in advertising, education, and entertainment.

Producing these videos traditionally is both time-consuming and expensive, requiring significant technical expertise. However, with advancements in AI-Generated Content (AIGC), automated video generation is becoming an increasingly researched area, offering the potential to streamline and enhance traditional video production processes. Techniques such as Text-to-Video (T2V) generation models (He et al., 2022; Ho et al., 2022; Singer et al., 2022; Zhou et al., 2022; Blattmann et al., 2023a; Chen et al., 2023a) and Image-to-Video (I2V) methods (Zhang et al., 2023a; Dai et al., 2023; Wang et al., 2024a; Zhang et al., 2023b) enable users to generate corresponding video outputs simply by inputting text or images.

While significant advancements have been made in video generation research, automating storytelling video production remains challenging. Current models struggle to preserve subject consistency throughout the complex process of storytelling video generation. Recent agent-driven systems, such as Mora (Yuan et al., 2024) and AesopAgent (Wang et al., 2024b), have been proposed to address Story-to-Video (S2V) generation by integrating multiple specialized agents, such as T2I and I2V generation agents. However, these methods fall short in allowing users to generate storytelling videos featuring their designated subjects, i.e., Customized Storytelling Video Generation (CSVG). The protagonists generated from story descriptions often exhibit inconsistency across multiple shots. Another line of research focusing on customized text-to-video generation like DreamVideo (Wei et al., 2023) and Magic-Me (Ma et al., 2024) can also be employed to synthesize storytelling videos. They first fine-tune the models using the data about the given reference protagonists, then generate the videos from the story descriptions. Despite these efforts, maintaining fidelity to the reference subjects remains a significant challenge. As shown in Figure 1, the results of TI-AnimateDiff, DreamVideo, and Magic-Me fail to preserve the appearance of the reference subject in the video. In these methods, the learned concept embeddings cannot fully capture and express the subject in different scenes.

Considering the limitations of existing storytelling video generation models, we explore the potential of multi-agent collaboration to synthesize customized storytelling videos. In this paper, we introduce a multi-agent framework called StoryAgent, which consists of multiple agents with distinct roles that work together to perform CSVG. Our framework decomposes CSVG into several subtasks, with each agent responsible for a specific role: 1) Story designer, writing detailed storylines and descriptions for each scene. 2) Storyboard generator, generating storyboards based on the story descriptions and the reference subject. 3) Video creator, creating videos from the storyboard. 4) Agent manager, coordinating the agents to ensure orderly workflow. 5) Observer, reviewing the results and providing feedback to the corresponding agent to improve outcomes. By leveraging the generative capabilities of different models, StoryAgent enhances control over the generation process, resulting in significantly improved character consistency. The core functionality of the agents in our framework can be flexibly replaced, enabling the framework to complete a wide range of video-generation tasks. This paper primarily focuses on the accomplishment of CSVG.

However, simply equipping the storyboard generator with existing T2I models, such as SDXL (Podell et al., 2023) as used by Mora and AesopAgent, often fails to preserve inter-shot consistency, i.e., maintaining the same appearance of customized protagonists across different storyboard images. Similarly, directly employing existing I2V methods such as SVD (Blattmann et al., 2023b) and Gen-2 (Esser et al., 2023) leads to issues with intra-shot consistency, failing to keep the character's fidelity within a single shot. Inspired by the image customization method AnyDoor (Chen et al., 2023b), we develop a new pipeline comprising three main steps—generation, removal, and redrawing—as the core functionality of the storyboard generator agent to produce highly consistent storyboards. To further improve intra-shot consistency, we propose a customized I2V method. This involves integrating a background-agnostic data augmentation module and a Low-Rank Adaptation with Block-wise Embeddings (LoRA-BE) into an existing I2V model (Xing et al., 2023) to enhance the preservation of protagonist consistency. Extensive experiments on both customized and public datasets demonstrate the superiority of our method in generating highly consistent customized storytelling videos compared to state-of-the-art customized video generation approaches. Readers can view the dynamic demo videos available at this anonymous link: `https://github.com/storyagent123/Comparison-of-storytelling-video-results/blob/main/demo/readme.md`[1]

The main contributions of this work are as follows: 1) We propose StoryAgent, a multi-agent framework for storytelling video production. This framework stands out for its structured yet flexible systems of agents, allowing users to perform a wide range of video generation tasks. These features

---

[1]The codes will be released upon the acceptance of the paper

also enable StoryAgent to be a prime instrument for pushing forward the boundaries of CSVG. 2) We introduce a customized Image-to-Video (I2V) method, LoRA-BE (Low-Rank Adaptation with Block-wise Embeddings), to enhance intra-shot temporal consistency, thereby improving the overall visual quality of storytelling videos. 3) In the experimental section, we present an evaluation protocol on public datasets for CSVG and also collect new subjects from the internet for testing. Extensive experiments have been carried out to prove the benefit of the proposed method.

## 2 RELATED WORK

**Story Visulization.** Our StoryAgent framework decomposes CSVG into three subtasks, including generating a storyboard from story descriptions, akin to story visualization. Recent advancements in Diffusion Models (DMs) have shifted focus from GAN-based (Li et al., 2019; Maharana et al., 2021) and VAE-based frameworks (Chen et al., 2022; Maharana et al., 2022) to DM-based approaches. AR-LDM (Pan et al., 2024) uses a DM framework to generate the current frame in an autoregressive manner, conditioned on historical captions and generated images. However, these methods struggle with diverse characters and scenes due to story-specific training on datasets like PororoSV (Li et al., 2019) and FlintstonesSV (Maharana and Bansal, 2021). For general story visualization, StoryGen (Chang Liu, 2024) iteratively synthesizes coherent image sequences using current captions and previous visual-language contexts. AutoStory (Wang et al., 2023) generates story images based on layout conditions by combining large language models and DMs. StoryDiffusion (Zhou et al., 2024) introduces a training-free Consistent Self-Attention module to enhance consistency among generated images in a zero-shot manner.Additionally, methods like T2I-Adapter (Mou et al., 2024), IP-Adapter (Ye et al., 2023), and Mix-of-Show (Gu et al., 2023), designed to enhance customizable subject generation, can also be used for storyboards. However, these often fail to maintain detail consistency across sequences. To address this, our storyboard generator, inspired by AnyDoor (Chen et al., 2023b), employs a pipeline of removal and redrawing to ensure high character consistency.

**Image Animation.** Animating a single image, a crucial aspect of storyboard animation, has garnered considerable attention. Previous studies have endeavored to animate various scenarios, including human faces (Geng et al., 2018; Wang et al., 2020; 2022), bodies (Blattmann et al., 2021; Karras et al., 2023; Siarohin et al., 2021; Weng et al., 2019), and natural dynamics (Holynski et al., 2021; Li et al., 2023; Mahapatra and Kulkarni, 2022). Some methods have employed optical flow to model motion and utilized warping techniques to generate future frames. However, this approach often yields distorted and unnatural results. Recent research in image animation has shifted towards diffusion models (Ho et al., 2020; Song et al., 2020; Rombach et al., 2022; Blattmann et al., 2023b) due to their potential to produce high-quality outcomes. Several approaches (Dai et al., 2023; Xing et al., 2023; Zhang et al., 2023c; Wang et al., 2024a; Zhang et al., 2023a) have been proposed to tackle open-domain image animation challenges, achieving remarkable performance for in-domain subjects. However, animating out-domain customized subjects remains challenging, often resulting in distorted video subjects. To address this issue, we propose LoRA-BE, aimed at enhancing customization generation capabilities.

**AI Agent.** Numerous sophisticated AI agents, rooted in large language models (LLMs), have emerged, showcasing remarkable abilities in task planning and utility usage. For instance, Generative Agents (Park et al., 2023) introduces an architecture that simulates believable human behavior, enabling agents to remember, retrieve, reflect, and interact. MetaGPT (Hong et al., 2024) models a software company with a group of agents, incorporating an executive feedback mechanism to enhance code generation quality. AutoGPT (Yang et al., 2023) and AutoGen (Wu et al., 2023) focus on interaction and cooperation among multiple agents for complex decision-making tasks. Inspired by these agent techniques, AesopAgent (Wang et al., 2024b) proposes an agent-driven evolutionary system for story-to-video production, involving script generation, image generation, and video assembly. While this method achieves consistent image generation, generating storytelling videos for customized subjects remains a challenge for AesopAgent.

## 3 STORYAGENT

As depicted in Figure 2, StoryAgent takes as inputs a prompt and a few videos of the reference subjects, and employs the collaborative efforts of five agents: the agent manager, story designer,

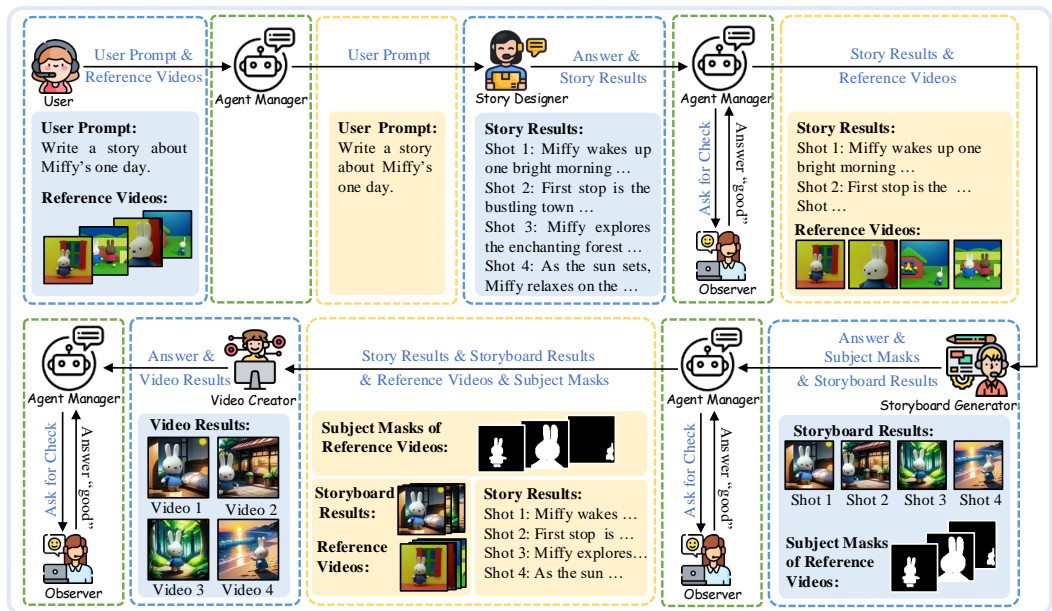

Figure 2: Our multi-agent framework's video creation process. Yellow blocks represent the next agent's input, while blue blocks indicate the current agent's output. For example, the Storyboard Generator (SG)'s input includes story results and reference videos, and its output consists of storyboard results and the subject mask of the reference videos. The Agent Manager (AM) automatically selects the next agent to execute upon receiving signals from different agents and may request the Observer to evaluate the results when other agents complete their tasks.

storyboard generator, video creator, and observer, to create highly consistent multi-shot storytelling videos. The workflow is segmented into three distinct steps: storyline generation, storyboard creation, and video generation.

During storyline generation, the agent manager forwards the user-provided prompt to the story designer, who crafts a suitable storyline and detailed descriptions $\mathbf{p} = \{p_1, \cdots, p_N\}$ (where $N$ represents the number of shots in the final storytelling video) outlining background scenes and protagonist actions. These results are then reviewed by the observer or user via the agent manager, and the process advances to the next step once the observer signals approval or the maximum chat rounds are reached.

The second step focuses on generating the storyboard $\mathbf{I} = \{I_1, \cdots, I_N\}$. Here, the agent manager provides the story descriptions $\mathbf{p}$ and protagonist videos $\mathbf{V}_{ref}$ to the storyboard generator, which produces a series of images aligned with $\mathbf{p}$ and $\mathbf{V}_{ref}$. Similar to the previous step, the storyboard results undergo user or observer evaluation until they meet the desired criteria. Finally, the story descriptions $\mathbf{p}$, storyboard $\mathbf{V}_{ref}$, and protagonist videos $\mathbf{V}_{ref}$ are handed over to the video creator for synthesizing multi-shot storytelling videos. Instead of directly employing existing models, as done by Mora, the storyboard generator and the video creator agents utilize a novel storyboard generation pipeline and the proposed LoRA-BE customized generation method respectively to enhance both inter-shot and intra-shot consistency. In the subsequent section, we will delve into the definitions and implementations of the agents within our framework.

## 3.1 LLM-BASED AGENTS

**Agent Manager.** Customized Storytelling Video Generation (CSVG) is a multifaceted task that necessitates the orchestration of several subtasks, each requiring the cooperation of multiple agents to ensure their successful completion in a predefined sequence. To facilitate this coordination, we introduce an agent manager tasked with overseeing the agents' activities and facilitating communication between them. Leveraging the capabilities of Large Language Models (LLM) such as GPT-4 (Achiam et al., 2023) and Llama (Touvron et al., 2023), the agent manager selects the next agent in line. This process involves presenting a prompt to the LLM, requesting the selection of the subsequent agent

Figure 3: The workflow diagrams of Storyboard Generator, along with the corresponding inputs (yellow blocks) and the outputs of their submodules (blue blocks).

from a predetermined list of available agents within the agent manager. The prompt, referred to as the role message, is accompanied by contextual information detailing which agents have completed their tasks. Empowered by the LLM's decision-making prowess, the agent manager ensures the orderly execution of tasks across various agents, thus streamlining the CSVG process.

**Story Designer.** In order to craft captivating storyboards and storytelling videos, crafting detailed, immersive, and narrative-rich story descriptions is crucial. To accomplish this, we introduce a story designer agent, which harnesses the capabilities of Large Language Models (LLM). This agent offers flexibility in LLM selection, accommodating models like GPT-4, Claude (Anthropic, 2024), and Gemini (Team et al., 2023). By prompting the LLM with a role message tailored to the story designer's specifications, including parameters such as the number of shots ($N$), background descriptions, and protagonist actions, the story designer generates a script comprising $n$ shots with corresponding story descriptions $\mathbf{p} = \{p_1, \cdots, p_n\}$, ensuring the inclusion of desired narrative elements.

**Observer.** The observer is an optional agent within the framework, and it acts as a critical evaluator, tasked with assessing the outputs of other agents, such as the storyboard generator, and signaling the agent manager to proceed or provide feedback for optimizing the results. At its core, this agent can utilize Aesthetic Quality Assessment (AQA) methods (Deng et al., 2017) or the general Multimodal Large Language Models (MLLMs), such as GPT-4 (Achiam et al., 2023) or LLaVA (Lin et al., 2023), capable of processing visual elements to score and determine their quality. However, existing MLLMs still have limited capability in evaluating images or videos. As demonstrated in our experiments in Appendix A.5, these models cannot distinguish between ground-truth and generated storyboards. Therefore, we implemented the LAION aesthetic predictor (Prabhudesai et al., 2024) as the core of this agent, which can effectively assess the quality of storyboards in certain cases and filter out some low-quality results. Nevertheless, current AQA methods remain unreliable. In practical applications, users have the option to replace this agent's function with human evaluation or omit it altogether to generate storytelling videos. Since designing a robust quality assessment model is beyond the scope of this paper, we will leave it for future work.

## 3.2 VISUAL AGENTS

**Storyboard Generator.** Storyboard generation requires maintaining the subject's consistency across shots. It is still a challenging task despite advancements in coherent image generation for storytelling (Wang et al., 2023; Zhou et al., 2024; Wang et al., 2024c) have been made. To address this, inspired by AnyDoor (Chen et al., 2023b), we propose a novel pipeline for storyboard generation that ensures subject consistency through removal and redrawing, as shown in Fig. 3. Initially, given detailed descriptions $\mathbf{p} = \{p_1, \cdots, p_N\}$, we employ text-to-image diffusion models like StoryDiffusion (Zhou et al., 2024) to generate an initial storyboard sequence $\mathbf{S} = \{s_1, \cdots, s_N\}$. During removal, each storyboard $s_n$ undergoes subject segmentation using algorithms like LangSAM, resulting in the subject mask $\mathbf{M} = \{m_1, \cdots, m_N\}$. For redrawing, a user-provided subject image with its background removed is selected, and StoryAnyDoor, fine-tuned based on AnyDoor with $\mathbf{V}_{ref}$, fills the mask locations $\mathbf{M}$ with the customized subject. Experiments in the following section prove that this strategy can effectively preserve the consistency of character details.

**Video Creator: LoRA-BE for Customized Image Animation.** Given the reference videos $\mathbf{V}_{ref}$, the storyboard $\mathbf{I}$, and the story descriptions $\mathbf{p}$, the goal of the video creator is to animate the storyboard following the story descriptions $\mathbf{p}$ to form the storytelling videos with consistent subjects of in $\mathbf{V}_{ref}$. Theoretically, existing I2V methods, such as SVD (Blattmann et al., 2023b), and SparseCtrl (Guo et al., 2023a), can equip the agent to perform this task. However, these methods still face significant

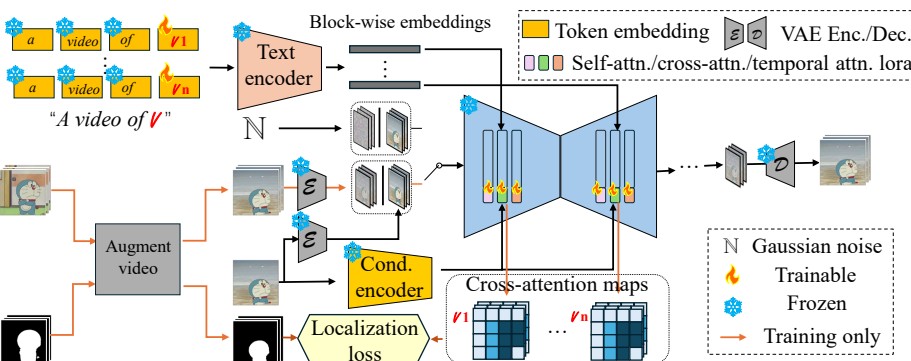

Figure 4: The illustration of our customized I2V generation method. Only the LoRA parameters inside each attention block and the block-wise token embeddings are trained to remember the subject. A localization loss is applied to enforce the tokens' cross-attention maps to focus on the subject.

challenges in maintaining protagonist consistency, especially when the given subject is a cartoon character like Miffy. Inspired by the customized generation concept in image domain, we propose a concept learning method, named LoRA-BE, to achieve customized I2V generation.

Our method is built upon a Latent Diffusion Model(LDM) (Ho et al., 2022)-based I2V generation model, DynamiCrafter(DC) (Xing et al., 2023). The modules in this method include a VAE encoder $\mathcal{E}_i$ and decoder $\mathcal{D}_i$, a text encoder $\mathcal{E}_T$, an image condition encoder $\mathcal{E}_c$, and a 3D U-Net architecture $\mathcal{U}$ with self-attention, temporal attention, and cross-attention blocks within. We first introduce the inference process of the valina DC. As shown in Figure 4, a noisy video $\mathbf{z}_T \in \mathbf{R}^{F \times C \times h \times w}$ is sampled from Gaussian distribution $\mathbb{N}$, where $F$ is the number of frames, and $C$, $h$, $w$ represent the channel dimension, height, and width of the frame latent codes. Then the condition image $I_n$, i.e., the storyboard in our task, is encoded by $\mathcal{E}$ and contacted with $\mathbf{z}_T$ as the input of U-Net $\mathcal{U}$. Additionally, the condition image is also projected by the condition encoder $\mathcal{E}_c$ to extract image embedding. Similar to the text embedding extracted by the text encoder from the text prompt $p_n$, the image embedding is injected into the video through the cross-attention block inside the U-Net. The output $\epsilon_T$ of U-Net will be used to denoise the noisy video $\mathbf{z}_T$ following the backward process $\mathcal{B}$ of LDM. The denoising process for the $n$-th shot at step $t$ can be written as:

$$\mathbf{z}_{t-1}^n = \mathcal{B}(\mathcal{U}([\mathbf{z}_t^n; \mathcal{E}_i(I_n)], \mathcal{E}_T(p_n), \mathcal{E}_c(I_n)), \mathbf{z}_t^n, t) \tag{1}$$

where $[\cdot; \cdot]$ means the concatenation operation along the channel dimension. We will drop off the subscript $n$ in the following content for simplicity.

Although the reference image is encoded to provide the visual information of the reference protagonist, the existing pre-trained DC model still fails to preserve the consistency of the out-domain subject. Hence, we propose to enhance its customization ability of animating out-domain subjects by fine-tuning. Inspired by the conclusions of Mix-of-Show (Gu et al., 2023) that fine-tuning the embedding of the new token, e.g., <Miffy>, helps to capture the in-domain subject, and fine-tuning to shift the pre-trained model, i.e., LoRA (Ryu, 2023), helps to capture out-domain identity, we enhance DC's customization ability from both aspects. Specifically, for each linear projection $L(x) = Wx$ in the self-attention, cross-attention, and temporal attention module, we add a few extra trainable parameters $A$ and $B$ to adjust the original projection to $L(x) = Wx + \Delta Wx = Wx + BAx$, thereby the generation domain of DC is shifted to the corresponding new subject after training. Moreover, we also train token embeddings for the new subject tokens. Unlike the Text Inversion (TI) method (Gal et al., 2022) which trains an embedding and injects the same embedding in all the cross-attention modules, we train different block-wise token embeddings. As there are 16 cross-attention modules in the U-Net, we add 16 new token embeddings $\mathbf{e} \in \mathbf{R}^{16 \times d}$, where $d$ represents the dimension of token embedding, for each new subject token, and each embedding is injected in only one cross-attention module. Consequently, to animate a new subject, only the LoRA parameters and 16 token embeddings are tuned to enhance the customized animation ability, where we use the given reference video $\mathbf{V}_{ref}$ to fine-tuning the model.

During training, the training sample $\mathbf{v} \in \mathbf{V}_{ref}$ is first projected into latent space by the AVE encoder $\mathbf{z}_0 = \mathcal{E}(\mathbf{v})$, then a noisy video is obtained by applying the forward process $\mathcal{F}$ of LDM on $\mathbf{z}_0$ with

Table 1: Comparison results of storytelling video generation on PororoSV and FlintstonesSV datasets.

| Dataset | Method | FVD ↓ | SSIM ↑ | PSNR ↑ | LPIPS↓ |
|---------|--------|-------|--------|--------|--------|
| PororoSV | SVD | 2634.01 | 0.5584 | 14.2813 | 0.3737 |
| | TI-Sparsectrl | 4209.80 | 0.5042 | 12.2749 | 0.5646 |
| | StoryAgent(ours) | **2070.56** | **0.6995** | **17.5104** | **0.2535** |
| FlintstonesSV | SVD | 1864.91 | 0.4460 | 14.5968 | 0.4023 |
| | TI-Sparsectrl | 3277.96 | 0.5571 | 14.7053 | 0.4958 |
| | StoryAgent(ours) | **991.37** | **0.6700** | **18.1169** | **0.2490** |

the sampled timestep $t$ and Gaussian noises $\epsilon \sim \mathbb{N}(0,1)$, $\mathbf{z}_t = \mathcal{F}(\mathbf{z}_0, t, \epsilon)$. The U-Net is trained to predict the noise $\hat{\epsilon}$ applied on $\mathbf{z}_0$, so that $\mathbf{z}_t$ can be recovered to $\mathbf{z}_0$ through the backward process. To reduce the interference of background information and make the trainable parameters focus on learning the identity of the new subject, we further introduce a localization loss $\mathcal{L}_{loc}$ applied on the across-attention maps. Specifically, the similarity map $D \in \mathbb{R}^{F \times h \times w}$ between the encoded subject token embedding and the latent videos is calculated for each cross-attention module, and the subject mask $m$ is leveraged to maximize the values of $D$ inside the subject locations. Hence, the overall training objective for the I2V generation can be formulated as follows:

$$\mathcal{L} = \mathcal{L}_{ldm} + \mathcal{L}_{loc} = \|\epsilon - \mathcal{U}([\mathbf{z}_t; \mathbf{z}_0[1]], \mathcal{E}_T(p), \mathcal{E}_c(\mathbf{v}[1]))\| - \frac{1}{F} \sum_f^F mean(D[f, m[f] = 1]) \quad (2)$$

As a result, the trainable subject embeddings and LoRA parameters can focus more on the subject.

## 4 EXPERIMENTS

**Implementation Details.** For storyboard generation, we employed AnyDoor as the redrawer and fine-tuned it to accommodate the new subject using the Adam optimizer with an initial learning rate of 1e-5. We selected 4-5 videos, each lasting 1-2 seconds, for every subject as reference videos, and conducted 20,000 fine-tuning steps. Regarding the training of the I2V model, we utilized DynamiCrafter (DC) (Xing et al., 2023) as the foundational model. We trained only the parameters of LoRA and block-wise token embeddings (LoRA-BE) using the Adam optimizer with a learning rate of 1e-4 for 400 epochs. All experiments were executed on an NVIDIA V100 GPU.

**Datasets and Metrics.** We employed two publicly available storytelling datasets, PororoSV (Li et al., 2019) and FlintstonesSV (Maharana and Bansal, 2021), which include both story scripts and corresponding videos, for evaluating our method. From PororoSV, we selected 5 characters, and from FlintstonesSV, we chose 4 characters as the customized subjects. For the training set, we selected reference videos for each subject from one episode, simulating practical application scenarios. For the testing set, we curated 10 samples for each subject, each consisting of 4 shots highly relevant to the subject. To evaluate our method on these datasets, we utilized reference-based metrics such as FVD (Unterthiner et al., 2018), PSNR, SSIM (Wang et al., 2004), and LPIPS (Zhang et al., 2018). Additionally, to assess the generalization ability, we collected 8 other subjects from YouTube and open-source online websites to form an open-domain set. Story descriptions for this set were generated using ChatGPT. Since there is no ground truth for this set, we reported the results on non-reference metrics as outlined in Liu et al. (2023), including Inception Score (IS), text-video consistency (Clip-score), semantic consistency (Clip-temp), Warping error, and Average flow (Flow-score). Arrows next to the metric names indicate whether higher (↑) or lower (↓) values are better for that particular metric. For Flow-Score, the arrow is replaced with a rightwards arrow (→) as it is a neutral metric.

### 4.1 EVALUATION ON PUBLIC DATASETS

**Quantitative Results.** The PororoSV (Li et al., 2019) and FlintstonesSV (Maharana and Bansal, 2021) datasets comprise story descriptions and corresponding videos, serving as ground truth for evaluating storytelling video generation methods. During testing, we generate a storyboard with a consistent background aligned with the ground-truth video. To achieve this, we use the first frame of each video with the subject removed as the initial storyboard. Subsequently, our storyboard generator redraws this initial storyboard to produce the final version. Finally, the generated storyboard is animated by the video creator agent to create a video of the subject.

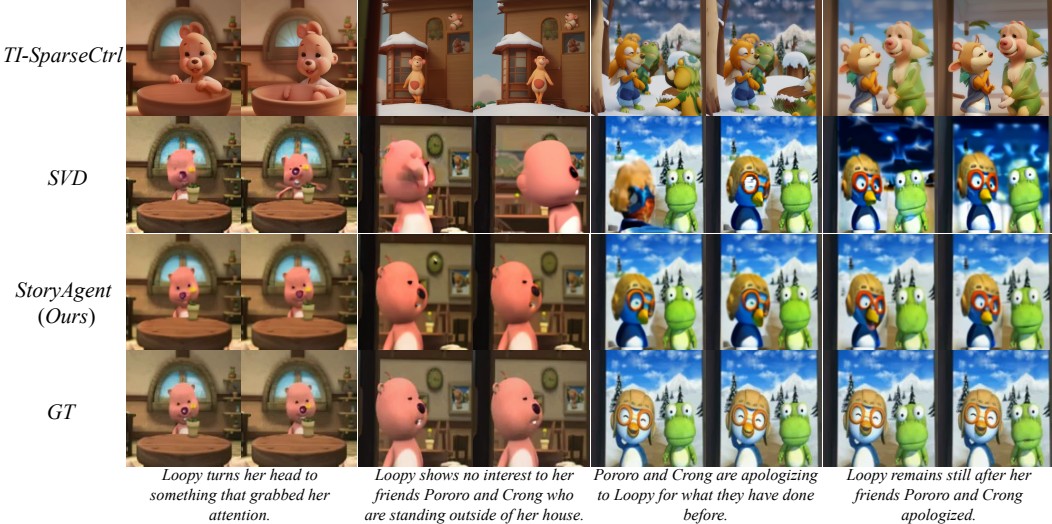

*Loopy turns her head to something that grabbed her attention.*    *Loopy shows no interest to her friends Pororo and Crong who are standing outside of her house.*    *Pororo and Crong are apologizing to Loopy for what they have done before.*    *Loopy remains still after her friends Pororo and Crong apologized.*

Figure 5: The Result visualization of three methods and the ground truth. The texts at the bottom are the story descriptions. The other two methods (the first 2 rows) fail to capture inter- and intra-shot consistency, our results (the $3_{rd}$ row) are more approaching the ground truth (the $4_{th}$ row).

In this evaluation framework, employing one-stage methods that directly generate storytelling videos from story descriptions yields significant discrepancies in the background compared to ground-truth videos. To ensure fair comparisons, we employ two I2V methods in conjunction with our storyboard generation as benchmarks: 1) SVD (Blattmann et al., 2023b), an open-source tool endorsed by recent work (Yuan et al., 2024) for image animation; 2) TI-SparseCtrl, wherein we augment the customization generation ability of SparseCtrl (Guo et al., 2023a) by integrating the Text Inversion (TI)(Gal et al., 2022) technique. Table 1 presents results computed against ground-truth videos. Our method consistently outperforms others by a notable margin across both video quality and human perception metrics, as evidenced by the FVD and LPIPS scores. Moreover, the improvement in the SSIM metric indicates closer alignment of our results with ground-truth videos, affirming the enhanced consistency of characters in our generated results.

**Qualitative Results.** To further elucidate the effectiveness of our approach, we qualitatively compare it with alternative methods in Figure 5. Our model demonstrates superior consistency compared to TI-SparseCtrl and SVD, closely resembling the ground truth. While TI-SparseCtrl, reliant on Text Inversion, struggles with maintaining consistency across shots, resulting in noticeable character variations, SVD manages to maintain inter-shot consistency but exhibits significant changes within shots, particularly evident in the $2_{nd}$ and $3_{rd}$ shots. Conversely, our method adeptly preserves both inter-shot and intra-shot consistency, thus affirming its effectiveness. Supplementary qualitative results are available in the Appendix.

## 4.2 EVALUATION ON OPEN-DOMAIN SUBJECTS

Table 2: Comparison results of storytelling video generation on the open-domain dataset.

| Method | Ours | TI-SparseCtrl | SVD | TI-AnimateDiff | DreamVideo | Magic-Me |
|---|---|---|---|---|---|---|
| IS ↑ | 2.6346 | 2.4184 | 2.3831 | 2.4539 | **3.4421** | 2.3989 |
| CLIP-score ↑ | **0.2053** | 0.1963 | 0.2013 | 0.2023 | 0.1843 | 0.2003 |
| CLIP-temp ↑ | 0.9985 | 0.9969 | 0.9959 | 0.9990 | 0.9963 | **0.9992** |
| Warping error ↓ | 0.0184 | 0.0189 | 0.0264 | **0.0043** | 0.0208 | 0.0048 |
| Flow-score → | 2.4332 | 2.6334 | 5.2117 | 1.8184 | 5.1140 | 1.4092 |

**Open-domain Dataset Results.** In this experiment, we also qualitatively compare our method with other CSVG methods, the video generation performance is shown in Figure 1. Due to the recent work, StoryDiffusion (Zhou et al., 2024), did not release the codes for video generation, we compare its storyboard generation performance in Figure 6. For other T2V methods, TI-AnimateDiff (Guo et al., 2023b), DreamVideo (Wei et al., 2023), and Magic-Me (Ma et al., 2024), we use the

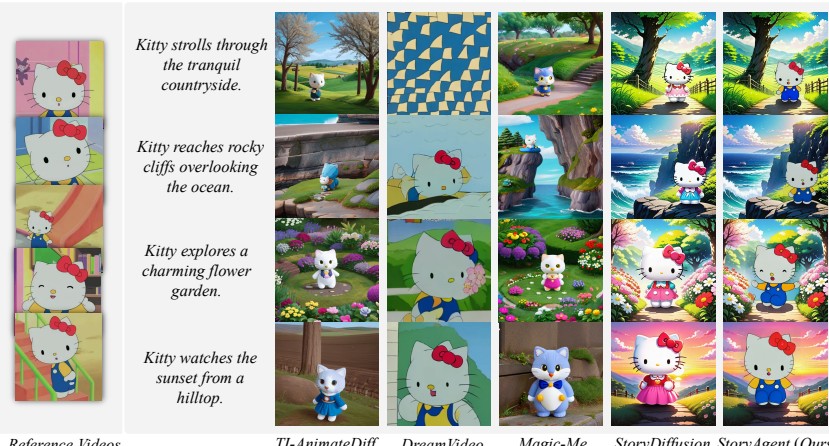

Figure 6: Storyboard generation visualization on open-domain subject (Kitty). The other four methods fail to preserve the consistency of the reference subject across shots, while our method effectively improves the consistency between the referenced image and the generated image.

first frames of the generated videos as the storyboard for comparison. As shown in Figure 1 and Figure 6, all these methods fail to capture inter-shot consistency. For the results of TI-AnimateDiff in Figure 6, the subject in the $3_{rd}$ shot is different from the subject in the $4_{th}$ shot. StoryDiffusion also cannot maintain the subject consistency across all shots. DreamVideo is unstable and produces unnatural content. Magic-Me even fails to maintain intra-shot subject consistency, as shown in the $4_{th}$ shot of Figure 1. More importantly, all these methods cannot preserve the reference subject in the generated videos. In contrast, our storyboard generator, based on the storyboard of StoryDiffusion, replaced the subjects with the reference subjects through the proposed removal and redrawing strategy. Compared with other methods, the proposed storyboard generation pipeline effectively preserves the consistency between the referenced image and the generated image in detail, such as the clothes of the subject, thereby enhancing the inter-shot consistency of the storytelling video. Besides, as proved by Figure 1, the video creator storing the subject information in a few trainable parameters further helps to maintain intra-shot consistency.

In addition to the subject consistency, we also report the quantitative results of all relevant methods, including TI-SparseCtrl and SVD using the storyboards from our agent, in Table 2. Our method outperforms other methods on text-video alignment while achieving comparable performances on other aspects like IS and semantic consistency (Clip-temp). These results indicate that our method can achieve high consistency while ensuring comparable video quality to other state-of-the-art methods. Therefore, the collaboration of multi-agents is a promising direction for achieving better results.

## 4.3 USER STUDIES

We conducted a user study on the results of different methods on the open-domain dataset and the Pororo dataset. We presented the results of different methods to the participants (They do not know which method each video comes from) and asked them to rate five aspects on a scale of 1-5: InteR-shot subject Consistency (IRC), IntrA-shot subject Consistency (IAC), Subject-Background Harmony (SBH), Text Alignment (TA) and Overall Quality (OQ). More details of the user studes can be seen in Appendix A.6.

Table 3: User studies of storytelling video generation on the open-domain dataset.

| Method | IRC ↑ | IAC ↑ | SBH ↑ | TA ↑ | OQ ↑ |
|---|---|---|---|---|---|
| TI- AnimateDiff | 2.9 | 3.8 | 3.4 | 2.7 | 3.0 |
| DreamVideo | 1.4 | 2.6 | 2.3 | 2.0 | 1.7 |
| Magic-me | 2.9 | 3.6 | 3.7 | 3.0 | 3.3 |
| TI-SparseCtrl | 2.6 | 2.4 | 2.9 | 2.8 | 2.5 |
| SVD | 3.4 | 3.0 | 3.4 | 2.8 | 2.8 |
| StoryAgent | **4.6** | **4.8** | **4.3** | **3.9** | **3.8** |

Table 4: User studies of storytelling video generation on the Pororo dataset.

| Method | IRC ↑ | IAC ↑ | SBH ↑ | TA ↑ | OQ ↑ |
|---|---|---|---|---|---|
| SVD | 3.5 | 2.9 | 3.4 | 3.4 | 3.1 |
| TI-SparseCtrl | 1.7 | 1.7 | 2.0 | 1.9 | 1.5 |
| LoRA-SparseCtrl | 2.5 | 2.1 | 2.0 | 2.0 | 1.9 |
| DC | 2.0 | 1.9 | 1.7 | 2.1 | 1.8 |
| LoRA-DC | 3.9 | 3.8 | 3.9 | 3.6 | 3.4 |
| StoryAgent | **4.8** | **4.8** | **4.5** | **4.3** | **4.4** |

Table 5: Ablation studies of video generation on PororoSV and FlintstonesSV datasets.

| Dataset | Method | FVD ↓ | SSIM ↑ | PSNR ↑ | LPIPS ↓ |
|---|---|---|---|---|---|
| PororoSV | DC-fintuening | 2251.47 | 0.4479 | 13.5322 | 0.4878 |
| | StoryAgent (ours) | **2070.56** | **0.6995** | **17.5104** | **0.2535** |
| FlintstonesSV | DC-finetuning | 3753.91 | 0.3357 | 10.4159 | 0.6042 |
| | StoryAgent(ours) | **991.37** | **0.6700** | **18.1169** | **0.2490** |

For the open-domain test, the methods evaluated included TI-AnimateDiff, DreamVideo, Magic-Me, TI-SparseCtrl, SVD, and our method StoryAgent. It is worth noting that SVD and TI-SparseCtrl are only video creators, so they used the storyboards generated by our Storyboard Generator as input. For the Pororo dataset, we used the ground-truth storyboard as input to evaluate the different Video Creator methods including SVD, TI-SparseCtrl, LoRA-SparseCtrl, Original DynamiCrafter (DC), LoRA-DC, Our StoryAgent. We have received 14 valid responses, and the average scores for each aspect are presented in Table 3 and Table 4. From the user studies conducted on the two datasets, it is evident that our method received the highest scores in all five evaluated aspects, especially the inter-shot consistency and the intra-shot consistency. This indicates that users prefer our method over others, demonstrating the superiority of our approach compared to existing methods.

### 4.4 ABLATION STUDIES

**Effectiveness of RoLA-BE.** One core contribution of this paper is the customized I2V generation. In this section, we will assess the results with and without this component. We finetuned the image injection module of DynamiCrafter (DC) (Xing et al., 2023) with the reference videos to improve the customization ability as the baseline. As shown in Table 5, without the proposed RoLA-BE, DC fails to preserve intra-shot consistency, and the score performance measuring the video quality and human perception decreases. The visualization results can be found in Appendix.A.4 In contrast, our method achieves better inter-shot and intra-shot consistency, while obtaining high-quality videos. These results suggest that the proposed method is effective in animating customized subjects.

### 5 CONCLUSION

We introduce StoryAgent, a multi-agent framework tailored for customized storytelling video generation. Recognizing the intricate nature of this task, we employ multiple agents to ensure the production of highly consistent video outputs. Unlike approaches that directly generate storytelling videos from story descriptions, StoryAgent divides the task into three distinct subtasks: story description generation, storyboard creation, and animation. Our storyboard generation method fortifies the inter-shot consistency of the reference subject, while the RoLA-BE strategy enhances intra-shot consistency during animation. Both qualitative and quantitative assessments affirm the superior consistency of the results generated by our StoryAgent framework.

**Limitations.** Although our method excels in maintaining consistency across character sequences, it faces challenges in generating customized human videos due to constraints in the underlying video generation model. Additionally, the duration of each shot remains relatively short. Moreover, limitations inherent in the pre-trained stable diffusion model constrain our ability to fully capture all text-specified details. One potential avenue for improvement involves training more generalized base models on larger datasets. Furthermore, enhancing our method to generate customized videos featuring multiple coherent subjects across multiple shots will be a primary focus of our future research. Further insights into the social impact of the proposed system are detailed in the Appendix.

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

# A  APPENDIX

The outline of the Appendix is as follows:

- More details of the agent scheduling process in Aagent Manager (AM).
- More evaluations on public datasets;
  - More storytelling video generation results on public datasets;
- More evaluations on open-domain subjects;
  - More storytelling video generation results on open-domain subjects;
- More ablation studies;
  - More storytelling video generation ablation on public datasets;
- The performance of Observer agent;
- The details of user studies.
- Social impact.

## A.1  MORE DETAILS OF THE AGENT SCHEDULING PROCESS IN AM

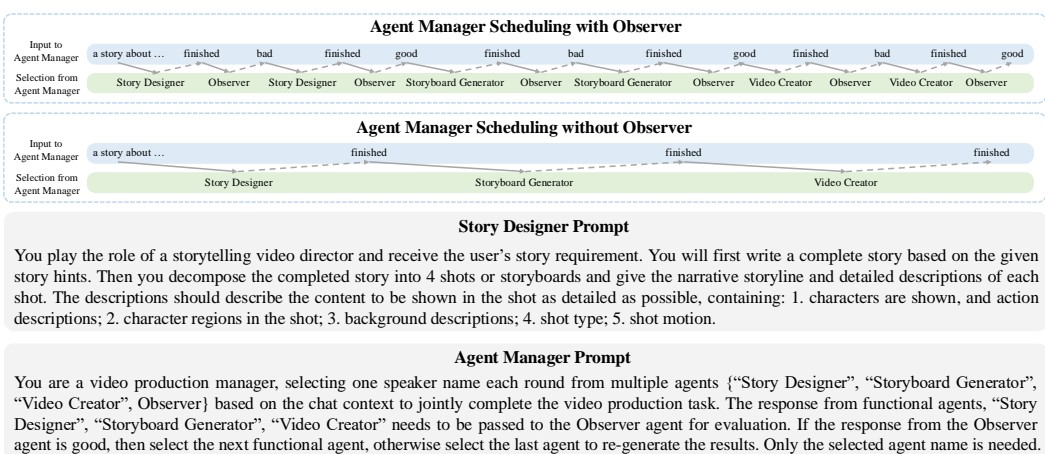

Figure 7: The agent scheduling process in AM. The solid arrows indicate AM's selection of an agent upon receiving a signal, while the dashed arrows represent the signals produced by the selected agent. Additionally, this figure shows the prompts used by the Story Designer and AM.

## A.2  MORE EVALUATIONS ON PUBLIC DATASETS

**More Storytelling Video Generation Results on Public Datasets.**

As mentioned before, existing I2V methods, such as SVD (Blattmann et al., 2023b), and SparseC-trl (Guo et al., 2023a), also can be used by our video creator to animate the storyboard $\mathbf{I}$ following the story descriptions $\mathbf{p}$ to form the storytelling videos. To further indicate the benefits of the proposed StoryAgent, we also visualize the storytelling videos generation results on FlintstonesSV dataset. As shown in Figure 8, our StoryAgent with the proposed LoRA-BE can not only generate results closer to the ground truth but also maintain the temporal consistency of subjects better, compared with the results generated by other methods.

## A.3  MORE EVALUATIONS ON OPEN-DOMAIN SUBJECTS

**More Storytelling Video Generation Results on Open-domain Subjects.**

Comparing our method with SVD (Blattmann et al., 2023b) and TI-SparseCtrl (Guo et al., 2023a), we also visualize more generated storytelling videos from story scripts on open-domain subjects, where the story descriptions are generated by our story designer agent. As shown in Figure 9 and

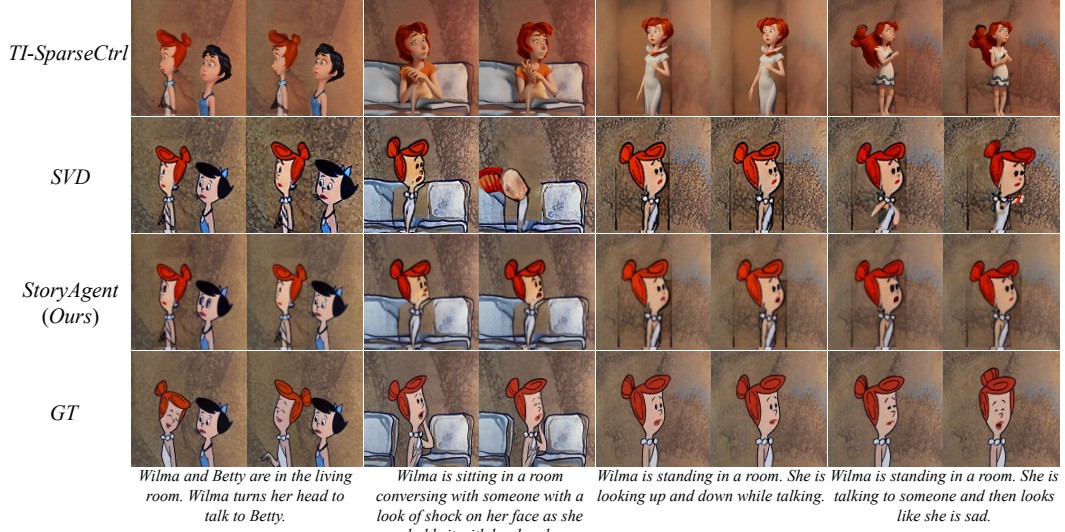

|  |  |  |  |
|---|---|---|---|
| *Wilma and Betty are in the living room. Wilma turns her head to talk to Betty.* | *Wilma is sitting in a room conversing with someone with a look of shock on her face as she holds it with her hand.* | *Wilma is standing in a room. She is looking up and down while talking.* | *Wilma is standing in a room. She is talking to someone and then looks like she is sad.* |

Figure 8: Storytelling videos generation visualization on FlintstonesSV dataset.

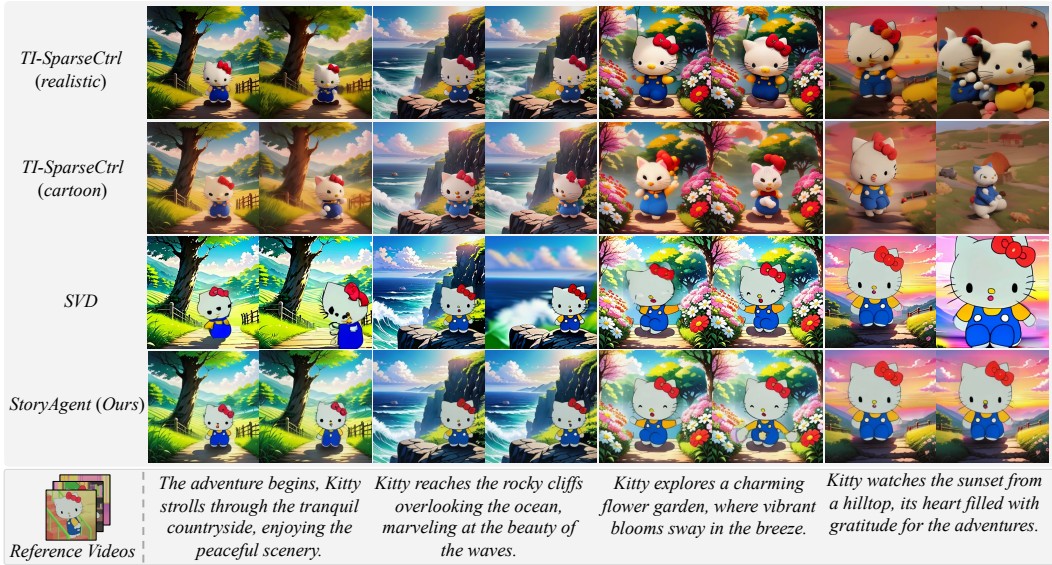

|  |  |  |  |
|---|---|---|---|
| *The adventure begins, Kitty strolls through the tranquil countryside, enjoying the peaceful scenery.* | *Kitty reaches the rocky cliffs overlooking the ocean, marveling at the beauty of the waves.* | *Kitty explores a charming flower garden, where vibrant blooms sway in the breeze.* | *Kitty watches the sunset from a hilltop, its heart filled with gratitude for the adventures.* |

Figure 9: Storytelling video generation visualization on open-domain subject (Kitty).

Figure 10, TI- SparseCtrl fails to maintain consistency throughout all the shots where the subjects change significantly in subsequent shots, such as the last shots on both of the two subjects. The proposed StoryAgent effectively maintain the temporal consistency between the referenced subjects throughout the story sequences in details, such as the clothes of cartoon subjects like Kitty and the appearance of real-world subjects like the bird. Although SVD also performs well in maintaining temporal consistency of the real-world bird in Figure 10, the movements of the bird are less able to follow the text, while our method can produce more vivid videos of the subject.

Furthermore, a comparison of an open-domain subject, a cartoon elephant, with state-of-the-art customization T2V methods is shown in Figure 11. It can be observed that TI-AnimatedDiff fails to capture inter-shot consistency, the subject in the $4_{th}$ shot is different from the subject in the $2_{nd}$ shot. DreamVideo occasionally falls short of generating the subject in the video. Magic-Me also fails to maintain inter-shot subject consistency. In contrast, our method can preserve the identity of the reference subject in all shots. These results further indicate that the storyboard generator agent in

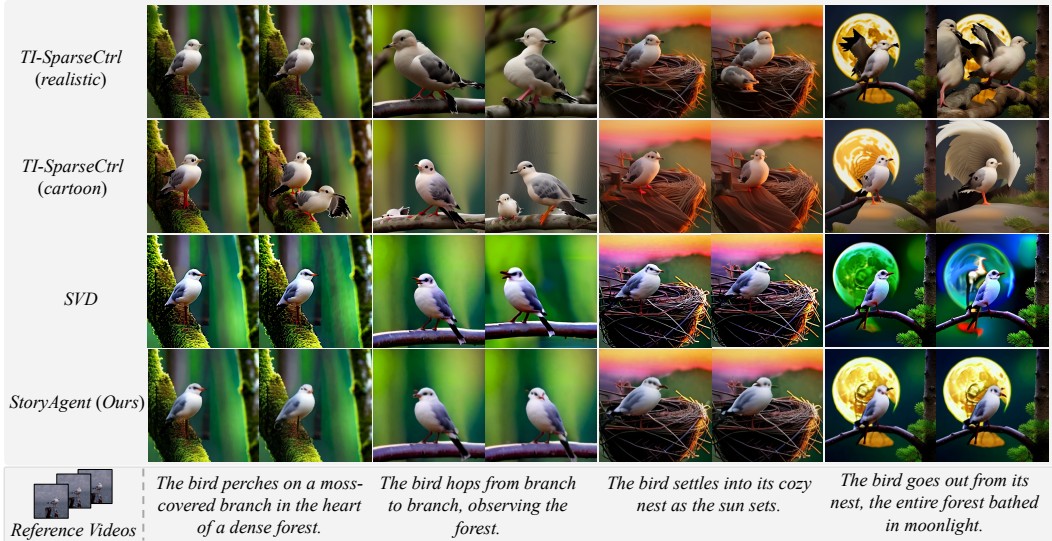

Figure 10: Storytelling video generation visualization on open-domain subject (The bird).

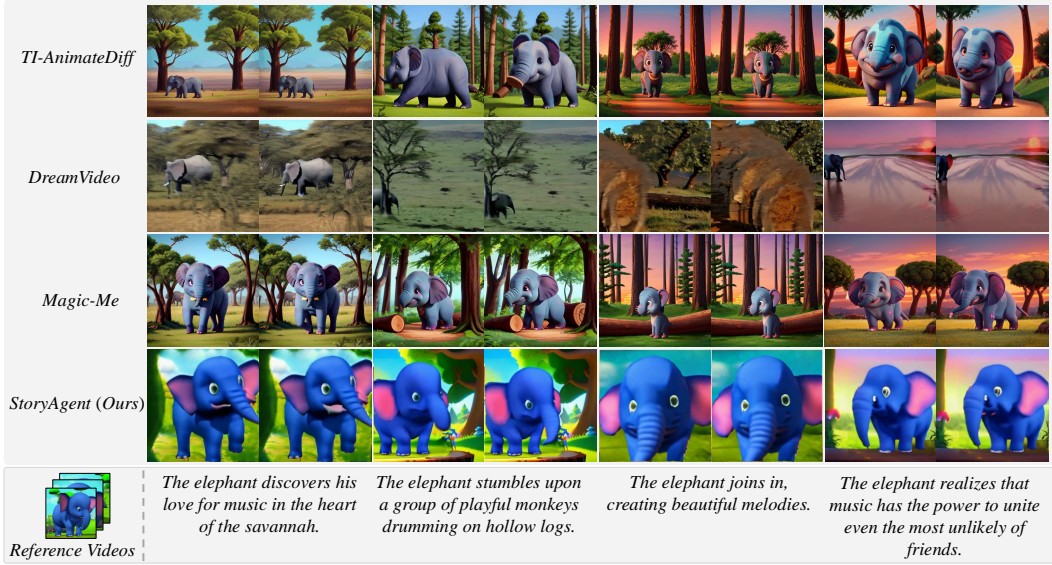

Figure 11: Storytelling video generation visualization on open-domain subject (The elephant). The other three methods (the 1-3 rows) fail to generate a consistent subject with the reference videos, while our method (the $4_{th}$ row) achieves high consistency.

our framework helps to improve the inter-shot consistency, and the video creator storing the subject information helps to maintain intra-shot consistency.

### A.4 MORE ABLATION STUDIES

**More Storytelling Video Generation Ablation on Public Datasets.**

The storytelling videos generation visualization on PororoSV dataset is also presented to further indicate the effectiveness of the proposed RoLA-BE. Same as the experimental settings in Section 4.4, we choose the finetuned DynamiCrafter (DC) (Xing et al., 2023) on the reference videos as the baseline, while our method consists of DC and the proposed RoLA-BE. As shown in Figure 12, DC still fails to generate customlized subjects even with the fine-tuning on the reference data, while our method generates results closer to the ground truth and fits the script well. Similarly, in Figure 13,

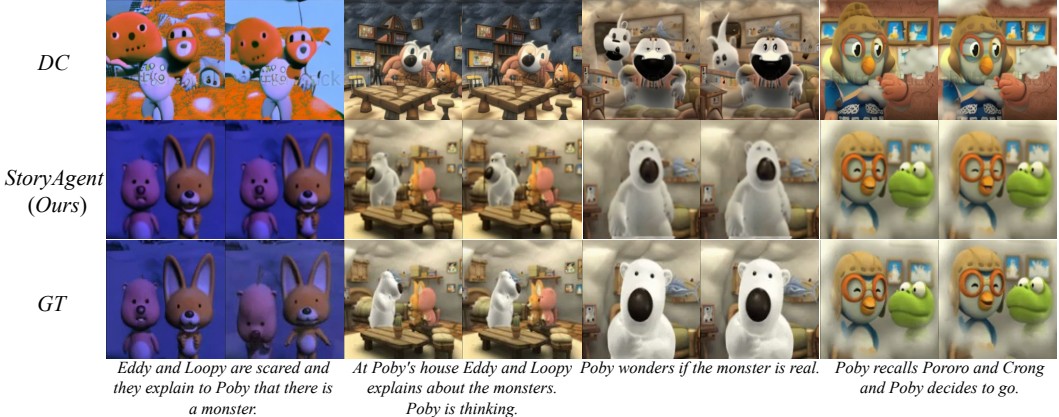

Figure 12: Storytelling videos generation ablation on PororoSV dataset.

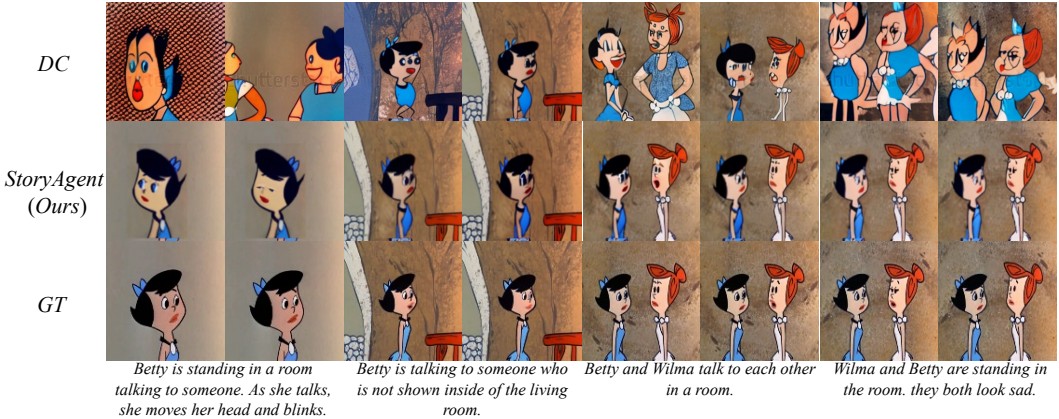

Figure 13: Storytelling videos generation ablation on FlintstonesSV dataset. Simply fine-tuning still results in inconsistency (the $1_{st}$ row), while our method (the $2_{nd}$ row) using the RoLA-BE strategy achieves more consistent results with the ground truth (the $3_{rd}$ row).

without the proposed RoLA-BE, DC fails to preserve intra-shot consistency (the $1_{st}$ row). In contrast, our method achieves better inter-shot and intra-shot consistency, while obtaining high-quality videos.

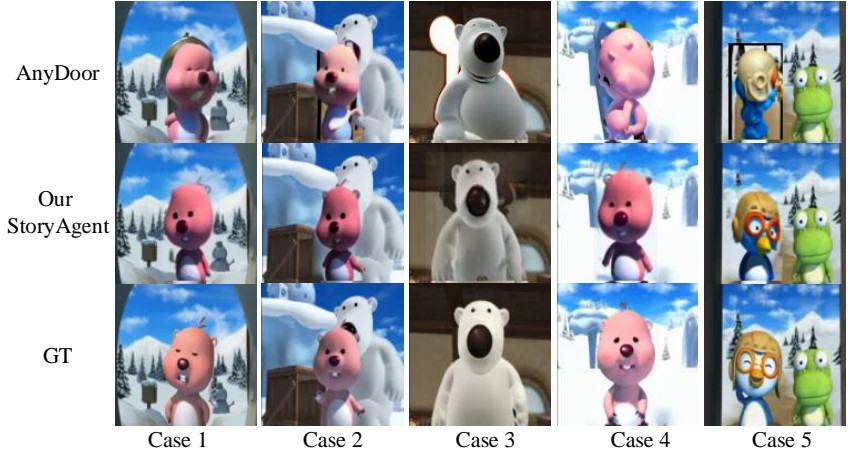

Figure 14: The case comparison of the Observer on the Pororo dataset.

Table 6: The score comparison of different observer functions on the Pororo dataset.

| Score model | Method | Case 1 | Case 2 | Case 3 | Case 4 | Case 5 |
|---|---|---|---|---|---|---|
| Gemini | AnyDoor | **8.0** | **8.0** | **8.0** | 7.0 | 5.0 |
| | Our StoryAgent | 7.0 | **8.0** | **8.0** | 7.0 | 5.0 |
| | GT | 5.0 | 4.0 | **8.0** | **9.0** | **9.0** |
| GPT-4o | AnyDoor | **6.0** | 4.0 | **4.0** | **4.5** | **3.5** |
| | Our StoryAgent | **6.0** | **4.5** | 3.5 | 3.5 | **3.5** |
| | GT | **6.0** | 4.0 | 3.5 | 3.5 | **3.5** |
| Aesthetic predictor | AnyDoor | 3.78 | 4.03 | 3.28 | **4.03** | 3.58 |
| | Our StoryAgent | 3.88 | **4.17** | 3.59 | 3.47 | 3.90 |
| | GT | **3.95** | 4.10 | **3.94** | 3.73 | **4.02** |

## A.5 THE PERFORMANCE OF OBSERVER

In this experiment, we use different aesthetic quality assessment methods, including two MultiModal Large Language Models, Gemini and GPT-4o, and the LAION aesthetic predictor V2 (Prabhudesai et al., 2024), to score the generated storyboards by the baseline methods Anydoor and our Storyboard Generator, and the ground-truth storyboard. The storyboard is shown in Figure 14, and the corresponding scores in the range of 1-10 are listed in Table 6.

We observed that MLLMs are not effective at distinguishing between storyboards of varying quality. For example, in case 4, GPT-4o assigns a high score to a low-quality result generated by AnyDoor, while giving the ground-truth image a lower score. Similarly, in case 2, Gemini exhibits the same behavior. Instead, the aesthetic predictor is relatively better at distinguishing lower-quality images, although it is still far from perfect. Therefore, in our experiments, we decided to bypass the observer agent to avoid wasting time on repeated generation. Further research on improving aesthetic quality assessment methods will be left for future work.

## A.6 THE DETAILS OF USER STUDIES

We conduct user evaluations by designing a comprehensive questionnaire to gather qualitative feedback. This questionnaire assesses five key indicators designed for personalized storytelling image and video generation:

(1) InteR-shot subject Consistency (IRC): Measures whether the features of the same subject are consistent among different shots (This indicator requires to consider the consistency of the subject among shots based on the provided subject reference images).

(2) IntrA-shot subject Consistency (IAC): Measures whether the features of the same subject are consistent in the same shot (This indicator only requires to consider the consistency of the subject in the same shot, without considering the subject reference images).

(3) Subject-Background Harmony (SBH): Measures whether the interaction between the subject and the background is natural and harmonious.

(4) Text Alignment (TA): Measure whether the video results match the textual description of the story.

(5) Overall Quality (OQ): Measures the overall quality of the generated storytelling videos.

The feedback collected will provide valuable insights to further refine our methods and ensure they meet the expectations of diverse audiences.

## A.7 SOCIAL IMPACT

Although storytelling video synthesis can be useful in applications such as education, and advertisement. Similar to general video synthesis techniques, these models are susceptible to misuse, exemplified by their potential for creating deep fakes. Besides, questions about ownership and copyright infringement may also arise. Nevertheless, employing forensic analysis and other manipulation detection methods could effectively alleviate such negative effects.

