# OpenReview forum: "StoryAgent: Customized Storytelling Video Generation via Multi-Agent Collaboration"
_ICLR.cc/2025/Conference — Submitted to ICLR 2025_

### Official Review · Reviewer_MwZc · 2024-11-02

**Soundness:** 3
**Presentation:** 3
**Contribution:** 3
**Rating:** 5
**Confidence:** 5

**Summary:**

This paper proposes StoryAgent for Customized Storytelling Video Generation (CSVG), which includes multiple agents for story design, storyboard generation, video creation, agent coordination, and result evaluation. The authors propose a pipeline for storyboard generation that ensures subject consistency through removal and redrawing, and a customized I2V generation method using LoRA to remember the subject. The authors conduct experiments on public datasets for CSVG and collect new subjects from the internet for testing.

**Strengths:**

1.	The paper is generally well-written and easy to follow, with clear and informative figures.
2.	In addition to evaluating public datasets such as PororoSV and FlintstonesSV, the authors also collected data from YouTube and various open-source websites to create an open-domain set, which can be used to evaluate Customized Storytelling Video Generation.
3.	Extensive experimental results demonstrate the effectiveness of the proposed framework for CSVG.

**Weaknesses:**

1.	My main concern with this paper is the lack of innovation. The authors utilize pre-trained large language models (LLMs) to generate story descriptions, then fine-tune the storyboard generator based on AnyDoor with reference videos to ensure subject consistency. They also use the well-established LoRA to fine-tune DynamiCrafter to remember new subjects. These separate agents do not sufficiently demonstrate innovation. Additionally, the authors adopt the simplest method to maintain character consistency, which is fine-tuning based on reference videos, rather than genuinely enhancing the model's out-of-domain generalization capabilities.
2.	The task setting of this paper seems a bit odd to me, as it requires providing a reference video for customized storytelling generation. Typically, users would only provide an image of a character. It appears to me that the authors adopted this setting primarily to fine-tune the storyboard generator and the I2V model with the reference video.
3.	Since the authors need to fine-tune StoryAnyDoor and DynamiCrafter for each new subject, they should provide the additional time required to fine-tune these modules in the paper to determine whether this approach is efficient.

**Questions:**

1.	In the video demonstrations provided by the authors, the videos generated by StoryAgent seem to have only slight movements compared to other methods. Is this due to the fine-tuning of DynamiCrafter with only reference videos?
2.	Why did the authors use the LAION aesthetic predictor as an observer, since it only evaluates the aesthetic quality of the generated images? This does not reflect the quality of the generated storyboards.

---

### Official Review · Reviewer_WU6E · 2024-11-03

**Soundness:** 3
**Presentation:** 3
**Contribution:** 3
**Rating:** 6
**Confidence:** 4

**Summary:**

This paper presents a new multi-agent framework to generate customized storytelling videos. The input to the system is a story generation prompt along with a few reference videos of the characters in the story. The proposed system comprises of a multiplicity of agents (agent manager, story designer, storyboard generator, video creator, observer). The proposed pipeline helps orchestrate the storytelling video generation in a manner that overcomes the character consistency problem of all existing works. The key contribution in this framework is the proposed LoRA-BE approach of generating videos from storyboard images that enforces consistency via finetuning of the DynamiCrafter model. Another contribution is a novel storyboard generation pipeline which enforces consistency via first generating a initial storyboard on which segmentation is done to obtain the subject (character) masks in every frame. Finally, the user provided images are employed for filling in the mask areas. As a result of these two contributions and the pipeline process that involves feedback of the observer agent for quality checks, the method is able to produce better quality storytelling videos as compared to the existing state of the art methods. Evaluation has been done on two existing data-sets with the standard evaluation metrics. In addition, a user study has been conducted that shows the superiority of the proposed approach.

**Strengths:**

(1) This paper provides a principled multi-agent based workflow for customized storytelling videos generation which mirrors the workflow of actual video production. This allows for agent-level methodological improvements as well as feedback for each stage of the generation process.

(2) The storyboard generation method introduces novelty via the generation, removal and redrawing steps that substantially improves the character consistency of the storyboarding frames. This critical step leads to better input for the video generation model.

(3) Incorporating the background-agnostic data augmentation and low-rank adapter into a image-to-video generation model helps generate much more consistent videos.

(40 The subjective evaluation results show the superiority of the proposed framework.

**Weaknesses:**

(1) Most of the components of the framework are incremental improvements to existing methods though the entire framework is novel.

(2) It is not clear in which of the results was the observer agent used and in which one the user feedback was utilized. This can a have a dramatic effect on the output quality. Can this be explicitly mentioned in the results tables? It would be useful to have one set of results using the automated observer and the human observer to see the impact of feedback quality on the output.

(3) The details of the insides of each agent has not been clearly described.

(4) The evaluation results in Table 2 do not quite convincingly support the conclusion that the proposed method has a substantial edge over other methods. It outperforms other methods only on one metric.

**Questions:**

(1)  Presumably the "customization" term in this paper refers to user-specified (visual) character (images) provided as input though it that has not been explicitly mentioned. Also, it is not very clear what is the substantive difference between "storytelling video generation" and "customized storytelling video generation". It would be very helpful if these terms are clearly defined in the introduction section of the paper.

(2) What are the requirements of the reference videos to be provided as input? It has not been articulated. Is it related to the amount of supervision signals provided to the various components of the workflow? It would be useful for the readers if the minimum length, the number of sample videos, the quality of the videos and the content requirements of the reference videos are clearly articulated. A discussion on how these factors affect the the different components as well as the final output quality will be useful.

(3) What is an "agent" in this paper? It is not at all clear. Is it merely a custom-prompt which is fed to a foundation model? Or is it something more with some degree of "agency" and autonomy. A clarification will be useful. What are the input to and output of each agent? And what degree of decision-making autonomy do they possess? It will be helpful to provide a clear definition of what constitutes an "agent" in the proposed framework, along with a table or diagram showing the inputs, outputs, and decision-making capabilities of each agent.

(4) For the entire pipeline of the approach, it is not clear which parts are fully automated and which parts need additional user-provided manual inputs. Especially given that the observer agents are shown to be substantially inferior to human users. Perhaps a table that clearly outlines which steps are automated and which require user input, along with an explanation of how the limitations of the observer agent are addressed in practice, can be added to the paper.

(5) What is "valina DC" on line 293? And what is "contacted with z_T" on line 296?

(6) In Table 1, it is surprising that SVD is better than T1-Spersectrl on the PororoSV Dataset while it is the other way around for the FlintstonesSV Dataset. What could be the explanation for this reversal?

(7) The user studies results in Table 3 and Table 4 are the strongest results for this work. However, the details of the study are missing such as the the number of subjects, subject demographics and the user study protocol. The protocol especially can have a dramatic effect on the results especially if it is not carefully designed. Therefore, it will be helpful to include more information concerning the number of participants, their demographics, the exact questions they were asked, how the videos were presented, and how potential biases were controlled for in the study design.

---

### Official Review · Reviewer_8CrA · 2024-11-04

**Soundness:** 2
**Presentation:** 2
**Contribution:** 2
**Rating:** 3
**Confidence:** 4

**Summary:**

This paper targets the task of storytelling video generation and proposes a multi-agent collaboration framework. Specifically, the pipeline includes agents for story design, a customized I2V, and Lora BE.
The input of the pipeline is a prompt for story description and some reference videos for describing the desired character. The output is generated storytelling videos with consistent characters.
The customized I2V is achieved based on DynamiCrafter, a general image-to-video pretrained model with Lora, blockwise token embeddings, and a localization loss. The story designer is achieved via LLMs such as GPT4, Claude, and Gemini.

**Strengths:**

The paper writing is generally clear.
The user study is provided.
The results show consistent characters across the clip.
The pipeline is automatic for storytelling generation.

**Weaknesses:**

Lack of comparisons and discussions with existing video storytelling methods:
- VideoStudio: Generating Consistent-Content and Multi-Scene Videos
- Vlogger: Vlogger: Make Your Dream A Vlog
- Animate-A-Story: Storytelling with Retrieval-Augmented Video Generation
- Anim-Director: A Large Multimodal Model Powered Agent for Controllable Animation Video Generation
Thus, we cannot tell the advantages and differences of this method compared with priors. The authors are suggested to add either generation results or discussions to highlight the differences and advantages of the proposed method.

Novelty
AI agent framework for storytelling video generation can be seen in previous work such as VideoStudio, Anim-Director, which makes this work less significant. The authors are suggested to highlight the unique value of this work compared to prior works.
There exist similar techniques like Blockwise embeddings, such as P+: Extended Textual Conditioning in Text-to-Image Generation, which expands the textual embeddings into different network layers.

Results
The generated storytelling videos lack meaningful motion and storyline. Instead, the results are some isolated video clips with difference scenes.

Grammer
line 395. "The Results"
Inconsistent figure reference: Fig. in line 258, but Figure in line 293.

Organization
There exists overlapped content between Figure 2 and Figure 3. A better figure organization is suggested.

**Questions:**

Evaluation
I'd like to know why the PSNR, SSIM, and LPIPS metrics are used for evaluating the generation task. Since these metrics are primarily used for reconstruction, for the generation task, the generated content is likely to have the same character but different actions, clothes, and scenes.

---

### Meta-Review · Area_Chair_BfxW · 2024-12-20

**Metareview:**

Paper was reviewed by three expert reviewers and received: 1 x reject, not good enough, 1 x marginally below the acceptance threshold, and 1 x marginally above the acceptance threshold ratings. Reviewers have raised a number of concerns with the work, including (1) limited or lacking novelty, as noted by all reviewers; (2) lacking exposition and description of the approach, and (3) issues with evaluation and task setting, among a few others. No rebuttal was provided to address these concerns. As a result, given the largely negative reviews and lack of rebuttal, AC is left with little choice but to recommend Rejection.

**Additional Comments On Reviewer Discussion:**

Discussion was not necessary, nor engaged in, since rebuttal was not provided.

---

### Decision · Program_Chairs · 2025-01-22

Reject